# Mobile phone MIMO antenna array miniaturization-based low SAR research in the combined EMF

Wen-Qi Hou [1], Yu-Xin Li[1], Ming-Fei Luo[1], Wen-Ying Zhou[1,2*], Mai Lu[1]

1 Key Laboratory of Opto-Electronic Technology and Intelligent Control of Ministry of Education, Lanzhou Jiaotong University, Lanzhou, China, 2 State Key Laboratory of Millimeter Waves, Nanjing, China

* zhouwy29@126.com

## Abstract

Due to the diversification of media functions of mobile phones, users can make calls and access the internet simultaneously, which has significantly increased the usage time of mobile phones. The exposure dose of the users in the combined electromagnetic fields (EMF) should be further quantified to better evaluate the public exposure safety. Different from most conventional EMF safety studies that only focus on a single frequency, this work not only discusses the mobile phone simultaneously operated in fourth-generation (4G) and fifth-generation (5G) mobile communications radiation impact on users, but also verifies that the miniaturized mobile phone multiple-input multiple-output (MIMO) antenna array can significantly reduce the specific absorption rate (SAR) absorbed by users. In this article, a miniaturized mobile phone MIMO antenna array is employed as the radiation source, and multi-pose human models are established to simulate the practical utilization of a smartphone. A systematic analysis of the SAR absorbed by the human model is conducted in both single and combined EMF scenarios. The results indicate that the peak SAR in various tissues under multi-frequency exposure is 1.02 to 15.85 times higher than that under single-frequency exposure.

## 1. Introduction

To meet the multi-standard communication requirements of 5G, Long-Term Evolution (LTE), Global System for Mobile Communications (GSM), Code Division Multiple Access (CDMA), and Wideband Code Division Multiple Access (WCDMA), mobile terminal antennas must support multi-band coverage, multi-channel radio frequency signal processing, and mode switching functions [1], while enabling users to simultaneously conduct calls and high-speed, large-capacity data transmission. Due to the convenience of mobile internet access, *The Statistical Report on China's Internet Development* shows that the average weekly online time per Chinese netizen

**Data availability statement:** All relevant data are within the paper.

**Funding:** This work was supported by the National Natural Science Foundation of China: [grant number 62161017, 61701208, 52467026]; The Department of Education of Gansu Province: [grant number 2024CXPT-11]; The Joint Research Foundation of Gansu Province: [grant number 24JRRA858]; Zhiyuan Laboratory: [grant number ZYL2024004]. The funders had no role in study design, data collection and analysis, decision to publish, or preparation of the manuscript.

**Competing interests:** The authors have declared that no competing interests exist.

has increased to 28.7 hours. Extended reliance on mobile devices subjects users to exposure to near-field radiation emitted by terminal antennas, especially when multiple functions are utilized concurrently. This combined EMF exposure has raised health concerns among both academia and the public [2]. Recent epidemiological surveys and experimental studies suggest that electromagnetic radiation from mobile devices can affect multiple physiological systems in humans [3,4]. Specifically, mobile phone radiation may influence cerebral cortex activity [5] and central nervous system function [6], and has been significantly associated with an increased risk of thyroid dysfunction [7,8]. Animal experimental studies also indicate that radiofrequency electromagnetic radiation can adversely affect organs and tissues such as the liver [9], heart [10], and lungs [11], thereby impairing their physiological functions.

Studies based on electromagnetic simulations indicate that the radiation levels of smartphones in typical usage scenarios are usually below the exposure limit of 2 W/kg set by the International Commission on Non-Ionizing Radiation Protection (ICNIRP) guideline [12]. However, SAR may exceed the safety limits when the exposure source is too close to the body [13,14]. Because the superposition of electromagnetic waves is complex and the corresponding biological effects are uncertain, the studies on combined EMF exposures are limited. Therefore, it is of great practical significance to research the influence of the combined EMF exposures from multi-frequency mobile phones on human tissues, and propose the corresponding electromagnetic protection measures.

Mobile phones could enhance the data transmission rate, reliability, and capacity of wireless communication systems by deploying multiple-input multiple-output (MIMO) antennas [15,16]. MIMO antenna miniaturization not only effectively saves space, enables multi-band coverage [17,18], but also directly decreases the area of induced currents in the near-field, thereby reducing radio frequency electromagnetic energy absorption by the user [19]. This study investigates the SAR in different body tissues under multiple postures when exposed to a miniaturized mobile phone MIMO antenna array. In addition, the effects of combined EMF exposure generated by antennas on human tissues when users use 4G and 5G networks simultaneously are discussed, with a focus on analyzing the differences between single-frequency and multi-frequency exposure. The findings of this study can offer a theoretical foundation and data backing for the optimized design of mobile terminal antennas and electromagnetic radiation protection.

## 2. Exposure source modeling and performance analysis

A miniaturized mobile phone MIMO antenna array is designed for the current mobile communication network environment [20]. The array antenna consists of 6 patch antennas (Ant1-Ant6) with FR-4 substrate (relative dielectric constant: 4.4), and the geometrical structure of the array antenna is shown in Fig 1. Ant1 and Ant2 use E-type radiating units (Fig 1c), which mainly cover the 2G and 5G bands. Ant3, Ant4, Ant5 and Ant6 use L-type radiating units (Fig 1d), which mainly cover the 3G and 4G bands. In order to reduce the coupling effect between the array elements during the miniaturization of this antenna array, the decoupling unit (Fig 1f) consisting of a

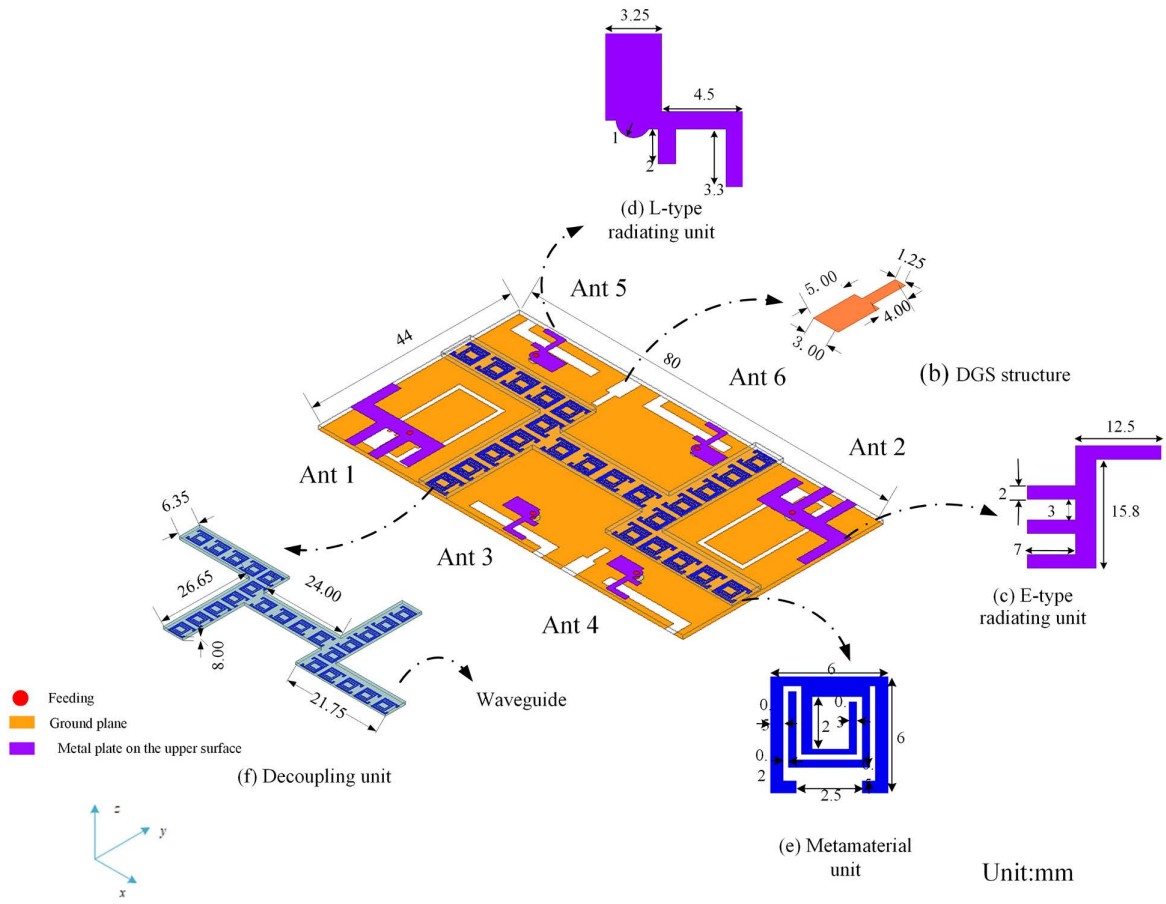

(d) L-type radiating unit

(b) DGS structure

(c) E-type radiating unit

(e) Metamaterial unit

(f) Decoupling unit

Feeding
Ground plane
Metal plate on the upper surface

Waveguide

Unit:mm

(a) Miniaturized MIMO mobile terminal antenna array

**Fig 1. Miniaturized mobile phone MIMO antenna array. (Unit: mm).**

metamaterial unit (Fig 1e) and a waveguide is added. The size of the antenna array is reduced by 49% compared to that without the decoupling unit.

The S-parameter simulation and measurement results of the miniaturized mobile phone MIMO antenna array are shown in Fig 2.

As shown in Fig 2a, the comparison results show that the coefficients of Ant 1 and Ant 2 ($S_{11}$, $S_{22}$) below −6 dB cover the range of 1.86 GHz~1.92 GHz and 3.41 GHz~3.58 GHz, which satisfy the 2G and 5G requirements. As shown in Fig 2b, the measured bandwidths of the reflection coefficients of Ant 3 and Ant 6 ($S_{33}$ and $S_{66}$) below −6 dB are 2.01 GHz~2.25 GHz and 2.02 GHz~2.25 GHz, respectively, satisfying the 3G frequency band (2.05 GHz-2.2 GHz) requirement. As shown in Fig 2c, the measurement bandwidths of the reflection coefficients of Ant 4 and Ant 5 ($S_{44}$, $S_{55}$) below −6 dB are 2.55 GHz~2.61 GHz and 2.55 GHz~2.61 GHz, satisfying the communication requirements of the 4G frequency band (2.59 GHz-2.66 GHz). Fig 2d-f shows the coupling coefficients ($S_{21}$, $S_{31}$, $S_{53}$, $S_{63}$, $S_{43}$, $S_{56}$) between pairs of radiating units, and their measured results are below −15 dB, which meet the design requirements.

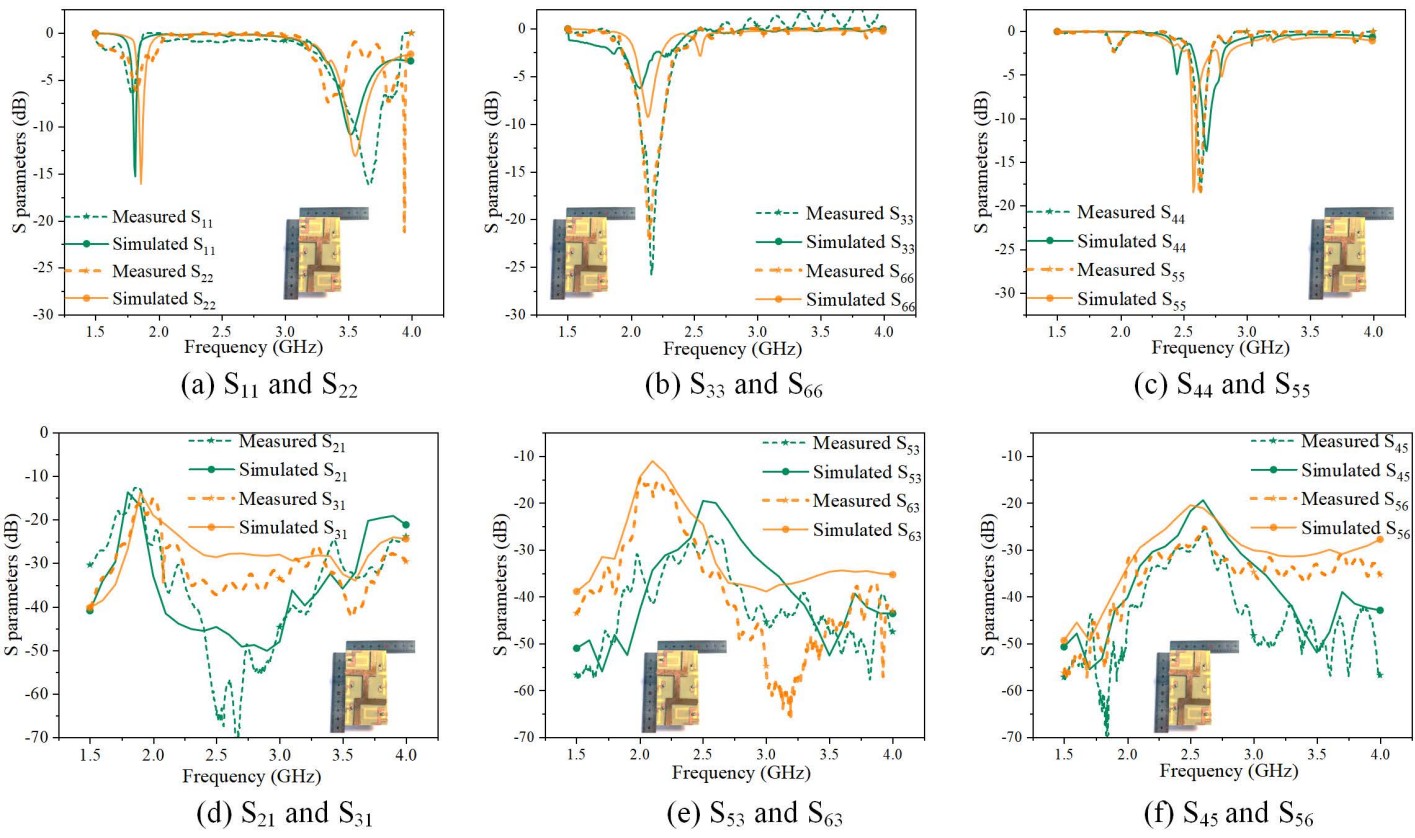

**Fig 2. S-parameter of the miniaturized mobile phone MIMO antenna array.**

The measurement results of the 2D radiation pattern of the miniaturized mobile phone MIMO antenna array are compared with the simulation results. The simulation results and the measured results of the normalized radiation pattern in the E-plane and H-plane at 1.9 GHz, 2.1 GHz, 2.6 GHz, and 3.5 GHz bands are shown in Fig 3.

Fig 3 shows that the antenna has good radiation performance in both E-plane and H-plane. The energy is concentrated in the endfire direction at frequencies of 1.9 GHz, 2.1 GHz, 2.6 GHz, and 3.5 GHz. The measured radiation pattern agrees well with the simulation results.

## 3. Electromagnetic exposure analyses of human with multi-pose

### 3.1. Near-field exposure modeling

Our previous research primarily focused on exploring an innovative approach to reduce radiation dose absorption within human head tissues [20]. In order to systematically analyze the near-field electromagnetic exposure effect of mobile phone MIMO antenna array on the human body, we construct electromagnetic simulation models for multi-pose, including calling pose, one-handed operation pose, and two-handed operation pose, as shown in Fig 4.

A human model with seven tissues is constructed, which includes the scalp, skull, brain, thyroid, heart, lungs, and liver, as shown in Fig 5.

We use the 4-Cole-Cole model to calculate the relative permittivity ($\varepsilon_r$) and the conductivity ($\sigma$) of seven human tissues at four frequencies (1.9 GHz, 2.1 GHz, 2.6 GHz, and 3.5 GHz), as shown in Table 1.

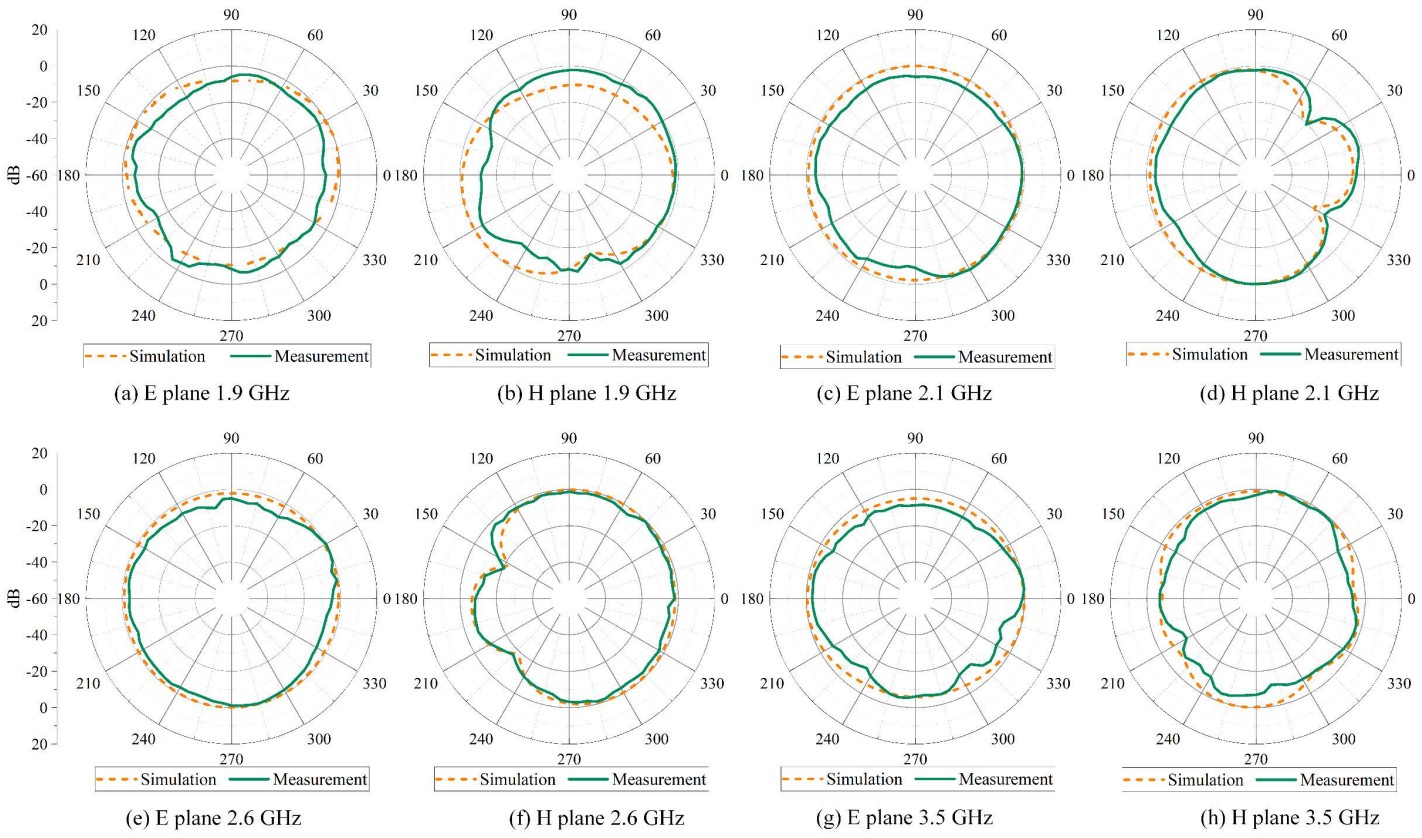

**Fig 3. Normalized radiation pattern of miniaturized mobile phone MIMO antenna array.**

### 3.2. Comparison of radiation doses absorbed by diverse tissues of the human body in the near-field at single frequency

For comparing the radiation dose absorbed by diverse tissues of the human model with multi-pose, we simulate and analyze the SAR values absorbed by diverse tissues when the mobile phone MIMO antenna array operates at frequencies of 1.9 GHz, 2.1 GHz, 2.6 GHz, and 3.5 GHz, respectively. The three different mobile terminal usage modes (calling pose, one-handed operation pose and two-handed operation pose) are considered as follows. To ensure compliance with current EMF safety standard, the peak SAR values are evaluated relative to the ICNIRP limit.

When users are calling with their mobile phones held in one hand, their heads and necks are closest to the exposure source. The SAR values absorbed by diverse tissues are shown in Table 2.

From Table 2, the results indicate that, in the calling posture, the SAR values absorbed by the hand's skin tissue are the highest and exhibit an increasing trend with the frequency. Due to the closer distance to the miniaturized MIMO array, the SAR values absorbed by the skull tissue and the thyroid tissue are larger than those of other tissues. As the distance between human tissues and the miniaturized mobile phone MIMO antenna array gradually increases, the Specific Absorption Rate (SAR) value presents a decreasing tendency at the same frequency.

When users operate their mobile phones with one hand, their bodies are closest to the exposure source. The SAR values absorbed by diverse tissues are shown in Table 3.

From Table 3, the results indicate that in the one-handed operation pose, the SAR values absorbed by the hand's skin tissue are maximal and exhibit an increasing trend with frequency. Owing to the relatively shorter distance to the

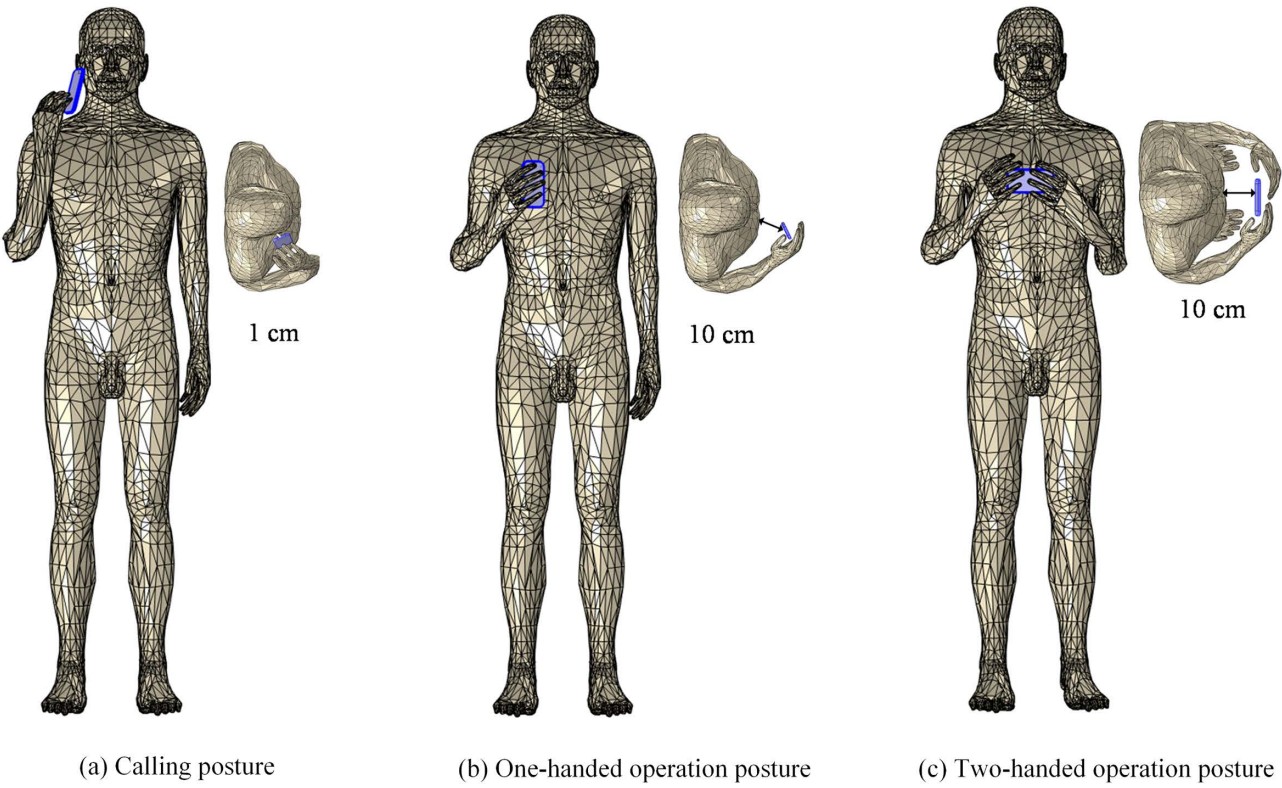

|     |     |     |
| --- | --- | --- |
| (a) Calling posture | (b) One-handed operation posture | (c) Two-handed operation posture |

**Fig 4. Multi-pose models for different mobile phone usage modes.**

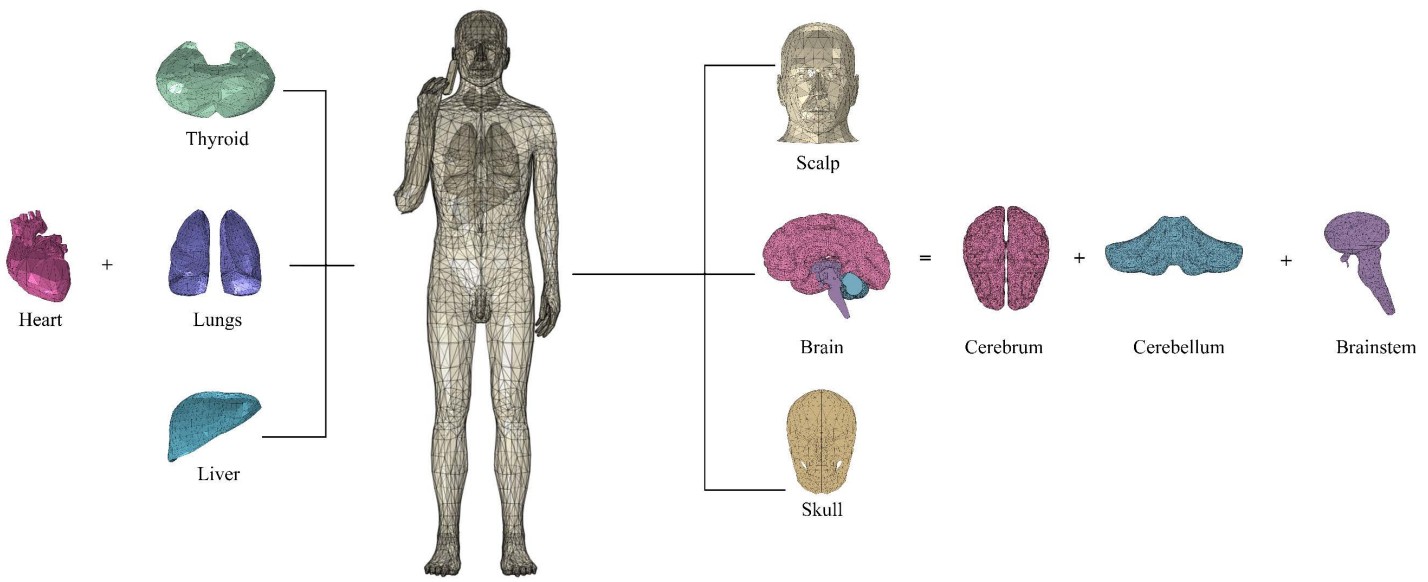

**Fig 5. Human model with seven core tissues.**

**Table 1. The dielectric parameters of seven human tissues at different frequencies.**

| Tissue | f=1.9 GHz | | f=2.1 GHz | | f=2.6 GHz | | f=3.5 GHz | | Tissue density kg·m⁻³ |
|---|---|---|---|---|---|---|---|---|---|
| | $\varepsilon_r$ | $\sigma$ (S/m) | $\varepsilon_r$ | $\sigma$ (S/m) | $\varepsilon_r$ | $\sigma$ (S/m) | $\varepsilon_r$ | $\sigma$ (S/m) | |
| Skin | 43.682 | 1.283 | 43.365 | 1.3897 | 42.645 | 1.684 | 41.473 | 2.308 | 1125 |
| Skull | 11.716 | 0.292 | 11.592 | 0.328 | 11.293 | 0.424 | 10.793 | 0.615 | 1990 |
| Brain | 36.868 | 0.958 | 36.600 | 1.047 | 35.991 | 1.292 | 35.003 | 1.810 | 1038 |
| Thyroid | 57.995 | 1.566 | 57.705 | 1.703 | 56.984 | 2.090 | 55.663 | 2.928 | 1250 |
| Heart | 56.063 | 1.841 | 55.579 | 1.985 | 54.508 | 2.380 | 52.831 | 3.205 | 1050 |
| Lung | 35.043 | 0.998 | 34.810 | 1.083 | 34.273 | 1.317 | 33.374 | 1.815 | 1151 |
| Liver | 44.012 | 1.346 | 43.638 | 1.464 | 42.790 | 1.788 | 41.417 | 2.468 | 1079 |

**Table 2. The peak SAR values of diverse tissues in calling pose (Unit: W/kg).**

| | 1.9 GHz | 2.1 GHz | 2.6 GHz | 3.5 GHz |
|---|---|---|---|---|
| Skin | 0.0339 | 0.0722 | 0.109 | 0.188 |
| Skull | $4.33 \times 10^{-4}$ | $7.07 \times 10^{-3}$ | 0.0173 | 0.0169 |
| Brain | $8.23 \times 10^{-5}$ | $7.05 \times 10^{-4}$ | $2.41 \times 10^{-3}$ | $5.73 \times 10^{-3}$ |
| Thyroid | $3.22 \times 10^{-4}$ | $1.89 \times 10^{-3}$ | 0.0155 | $2.73 \times 10^{-3}$ |
| Heart | $3.88 \times 10^{-6}$ | $4.70 \times 10^{-5}$ | $3.50 \times 10^{-5}$ | $3.93 \times 10^{-6}$ |
| Lungs | $1.56 \times 10^{-5}$ | $2.09 \times 10^{-4}$ | $8.25 \times 10^{-5}$ | $1.19 \times 10^{-4}$ |
| Liver | $6.58 \times 10^{-6}$ | $7.64 \times 10^{-5}$ | $7.86 \times 10^{-5}$ | $4.30 \times 10^{-4}$ |

**Table 3. The peak SAR values of diverse tissues in one-handed operation pose (Unit: W/kg).**

| | 1.9 GHz | 2.1 GHz | 2.6 GHz | 3.5 GHz |
|---|---|---|---|---|
| Skin | 0.0149 | 0.0677 | 0.132 | 0.180 |
| Skull | $7.36 \times 10^{-5}$ | $6.80 \times 10^{-4}$ | $1.74 \times 10^{-3}$ | $7.92 \times 10^{-4}$ |
| Brain | $1.58 \times 10^{-5}$ | $3.82 \times 10^{-4}$ | $1.02 \times 10^{-3}$ | $5.19 \times 10^{-4}$ |
| Thyroid | $3.16 \times 10^{-5}$ | $2.60 \times 10^{-4}$ | $1.94 \times 10^{-3}$ | $8.76 \times 10^{-4}$ |
| Heart | $5.59 \times 10^{-5}$ | $4.71 \times 10^{-4}$ | $1.65 \times 10^{-4}$ | $2.07 \times 10^{-5}$ |
| Lungs | $1.43 \times 10^{-4}$ | $1.75 \times 10^{-3}$ | $1.07 \times 10^{-3}$ | $3.90 \times 10^{-4}$ |
| Liver | $1.45 \times 10^{-4}$ | $7.71 \times 10^{-4}$ | $1.81 \times 10^{-3}$ | $7.91 \times 10^{-4}$ |

miniaturized MIMO terminal antenna array, the SAR values absorbed by the heart, lungs, and liver increase by over 10 times in comparison to the calling posture. Conversely, the SAR values absorbed by the skull, brain, and thyroid decrease by a factor of 10. The peak SAR values of the skull, brain, and liver all occur at a frequency of 2.6 GHz, whereas those of the heart and lungs occur at a frequency of 2.1 GHz.

When users operate their mobile phones with two hands, the exposure source is positioned at the center in front of the chest. The SAR values absorbed by diverse tissues are shown in Table 4.

From Table 4, the results indicate that, in the two-handed operation pose, the SAR values absorbed by the skin tissue of the hand are the largest. Owing to the fact that the MIMO antenna array is located at the central position in front of the chest, the distribution pattern of each tissue bears resemblance to that in the one-handed operation posture. Nevertheless, when compared to the one-handed operation pose, the SAR in the heart exhibits an increase, whereas the SAR in the lungs shows a decrease.

**Table 4. The peak SAR values of diverse tissues in two-handed operation pose (Unit: W/kg).**

|          | 1.9 GHz | 2.1 GHz | 2.6 GHz | 3.5 GHz |
|----------|---------|---------|---------|---------|
| Skin     | 0.0449 | 0.0504 | 0.0726 | 0.123 |
| Skull    | $9.51 \times 10^{-4}$ | $4.35 \times 10^{-4}$ | $9.07 \times 10^{-4}$ | $7.23 \times 10^{-4}$ |
| Brain    | $2.66 \times 10^{-4}$ | $1.69 \times 10^{-4}$ | $3.91 \times 10^{-4}$ | $4.55 \times 10^{-4}$ |
| Thyroid  | $4.82 \times 10^{-5}$ | $6.19 \times 10^{-5}$ | $4.52 \times 10^{-4}$ | $2.65 \times 10^{-4}$ |
| Heart    | $4.37 \times 10^{-4}$ | $5.92 \times 10^{-4}$ | $2.91 \times 10^{-4}$ | $5.31 \times 10^{-5}$ |
| Lungs    | $7.34 \times 10^{-4}$ | $7.51 \times 10^{-4}$ | $9.08 \times 10^{-4}$ | $6.12 \times 10^{-4}$ |
| Liver    | $3.42 \times 10^{-4}$ | $1.53 \times 10^{-4}$ | $6.72 \times 10^{-4}$ | $8.08 \times 10^{-4}$ |

The miniaturized mobile phone MIMO antenna array put forward in this paper is compared with the peak SAR in near-field electromagnetic exposure in the existing literature, as presented in Table 5.

Specifically, Table 5 illustrates that under near-field exposure scenarios, the higher the operating frequency of the antenna, the greater the radiated EMF energy, which in turn leads to a higher SAR in tissues close to the exposure source. When antennas operate at the same frequency, a more compact antenna size corresponds to less radiated energy, resulting in a lower SAR in tissues close to the exposure source. Compared to other designs in the literature, the proposed antenna exhibits advantages of a lower profile, more compact size, and minimal radiation impact on human tissues.

The above results show the radiation impact when users operate a single function of mobile phone MIMO antenna array, with all peak SAR values under single-frequency exposure remaining below the ICNIRP safety limit of 2 W/kg. However, current smart mobile terminals can also perform multimedia data interaction during calls, making it necessary to analyze the combined EMF exposure when the mobile terminal operates at two working frequencies simultaneously.

## 4. SAR comparison between single and combined EMF exposure

As smartphones are widely used, voice calls and data transmission frequently occur simultaneously, thereby exposing the human body to combined EMF of different frequencies. Existing laboratory studies on rodents include the following: In Ref.[27], the accumulative effects of microwave radiation on cognitive function were investigated by sequentially exposing rats to 2.856 GHz and 1.5 GHz microwaves. The results showed that the microwave-induced cognitive decline was largely determined by the power density rather than the frequency. Meanwhile, Ref. [28] examined the effects of single and combined high-power microwave exposure on male reproduction by simultaneously exposing rats to 1.5 GHz and 4.3

**Table 5. Comparison of peak SAR values across different antenna.**

| Refs. | Antenna Dimensions | Frequency | Maximum SAR Value |
|-------|--------------------|-----------|-------------------|
| [12] | $13.65 \times 6.94 \times 0.95\ cm^3$ | 5.2 GHz | 1.77 W/kg |
| [21] | $38 \times 83 \times 1.524\ mm^3$ | 4.8 GHz | 1.17 W/kg |
| [22] | $24 \times 24 \times 1.6\ mm^3$ | 3.6 GHz | 1.80 W/kg |
| [23] | $140 \times 70 \times 1\ mm^3$ | 5.5 GHz | 1.05 W/kg |
| [24] | $150 \times 75 \times 0.8\ mm^3$ | 5.3 GHz | 1.31 W/kg |
| [25] | $50 \times 10 \times 0.8\ mm^3$ | 3.8 GHz | 0.52 W/kg |
| [26] | $75 \times 150 \times 1.6\ mm^3$ | 3.5 GHz | 1.85 W/kg |
| Prop. Work | $44 \times 80 \times 0.8\ mm^3$ | 3.5 GHz | 0.19 W/kg |

GHz microwaves. Regarding simulation studies on human models, Ref [29] analyzed the SAR in a numerical human head model to simulate real smartphone use. The total SAR was obtained by summing the values calculated for each frequency component. However, this study did not take into account the permittivity and conductivity impact for combined SAR value. Furthermore, the SAR calculation was limited to the head surface with an idealized radiated power, and other tissues were not considered.

Ref. [30] summed the SAR values from 3G, 4G, and 5G center-frequency waves under x-, y-, and z-axis polarizations.

The SAR distributions at the two frequencies combined EMF are used for an integrative dosimetric evaluation in the human tissue exposures, i.e., the SAR at 2.6 GHz for 4G, the SAR at 3.5 GHz for 5G. The following integrative SAR distribution is obtained.

$$SAR_{integrative}(x,y,z) = S\hat{A}R_{f_1}(x,y,z) \cdot \sum_{n=1}^{i} E_{f_1,n}^2 + \cdots S\hat{A}R_{fz}(x,y,z) \cdot \sum_{n=1}^{i} E_{fz,n}^2 \tag{1}$$

The exposure source operates at frequency point $f$, and the three upward components of the emitted electromagnetic wave in three-dimensional coordinates $(x,y,z)$ are equated to an incident plane wave with an electric field strength of 1 V/m. $S\hat{A}R_f(x,y,z)$ is the normalized value of the absorbed radiation dose to human tissues under the radiation from this exposure source, but it is unclear whether the permittivity and conductivity of the SAR at different frequencies are taken into account.

Based on the above studies, we further analyze the distribution of SAR in vital tissues of the human body when the user simultaneously uses the call function of 4G frequency band (center frequency: 2.6 GHz) and a data transmission function of 5G frequency band (center frequency: 3.5 GHz). Firstly, we determine the normalized values of the electric field strength in the three directions of the three dimensional coordinates $(x,y,z)$ when the exposed source operates at the above two frequencies. Then, we calculate the maximum value of the electric field strength when the phase $\psi$ varies from 0 to 360 degrees. The maximum value of the electric field strength $\left|\hat{E}\right|$ is calculated as shown in Equation 2:

$$|\hat{E}|_{2.6GHz} = \hat{E}_{x(2.6GHz)}sin\theta cos\psi + \hat{E}_{y(2.6GHz)}sin\theta cos\psi + \hat{E}_{z(2.6GHz)}sin\theta cos\psi$$
$$|\hat{E}|_{3.5GHz} = \hat{E}_{x(3.5GHz)}sin\theta cos\psi + \hat{E}_{y(3.5GHz)}sin\theta cos\psi + \hat{E}_{z(3.5GHz)}sin\theta cos\psi \tag{2}$$

where $\theta$ is the angle of incidence of the electromagnetic wave.

Next, the combined exposure $SAR_{combined}$ in diverse tissues from two frequencies is assessed based on Equation 3 of the ICNIRP guideline [31].

$$SAR = \frac{\int_v \sigma_{2.6GHz}|E|_{2.6GHz}^2 dv + \int_v \sigma_{3.5GHz}|E|_{3.5GHz}^2 dv}{\int_v \rho dv} \tag{3}$$

where $V = \int_v dv$ is the volume of the integral.

Each permittivity and conductivity of each human tissue correspond to a single operating frequency of the mobile phone, where $V = \int_v dv$ is the volume of the integral. The permittivity and conductivity of every human tissue are characteristic of each operating frequency of the mobile phone.

Based on the above theoretical calculation, we analyze the SAR distribution in diverse tissues of the human model under the combined EMF environment at both 2.6 GHz and 3.5 GHz by using COMSOL Multiphysics. The input power of each antenna is 0.05 W. Then, we compare the multi-frequency exposure results with the single-frequency exposure results, with the maximum values shown in Table 6 and the distributional differences shown in Figs 6-12.

**Table 6. Comparison of the peak SAR values across different scenarios (Unit: W/kg).**

| Tissue | Usage Mode | Single-frequency SAR | | Multi-frequency SAR | ICNIRP General Public Restriction SAR |
|---|---|---|---|---|---|
| | | 2.6 GHz | 3.5 GHz | 2.6 + 3.5 GHz | |
| Skin | Calling | 0.109 | 0.188 | 0.343 | 2.000 |
| | One-handed | 0.132 | 0.180 | 0.193 | |
| | Two-handed | 0.0726 | 0.123 | 0.131 | |
| Skull | Calling | 0.0173 | 0.0169 | 0.0244 | |
| | One-handed | $1.74 \times 10^{-3}$ | $7.92 \times 10^{-4}$ | $2.41 \times 10^{-3}$ | |
| | Two-handed | $9.07 \times 10^{-4}$ | $7.23 \times 10^{-4}$ | $1.53 \times 10^{-3}$ | |
| Brain | Calling | $2.41 \times 10^{-3}$ | $5.73 \times 10^{-3}$ | $7.05 \times 10^{-3}$ | |
| | One-handed | $1.02 \times 10^{-3}$ | $5.19 \times 10^{-4}$ | $1.23 \times 10^{-3}$ | |
| | Two-handed | $3.91 \times 10^{-4}$ | $4.55 \times 10^{-4}$ | $7.56 \times 10^{-4}$ | |
| Thyroid | Calling | 0.0155 | $2.73 \times 10^{-3}$ | 0.0156 | |
| | One-handed | $1.94 \times 10^{-3}$ | $8.76 \times 10^{-4}$ | $2.08 \times 10^{-3}$ | |
| | Two-handed | $4.52 \times 10^{-4}$ | $2.65 \times 10^{-4}$ | $4.57 \times 10^{-4}$ | |
| Heart | Calling | $3.50 \times 10^{-5}$ | $3.93 \times 10^{-6}$ | $3.67 \times 10^{-5}$ | |
| | One-handed | $1.65 \times 10^{-4}$ | $2.07 \times 10^{-5}$ | $1.77 \times 10^{-4}$ | |
| | Two-handed | $2.91 \times 10^{-4}$ | $5.31 \times 10^{-5}$ | $3.12 \times 10^{-4}$ | |
| Lung | Calling | $8.25 \times 10^{-5}$ | $1.19 \times 10^{-4}$ | $1.98 \times 10^{-4}$ | |
| | One-handed | $1.07 \times 10^{-3}$ | $3.90 \times 10^{-4}$ | $1.16 \times 10^{-3}$ | |
| | Two-handed | $9.08 \times 10^{-4}$ | $6.12 \times 10^{-4}$ | $1.17 \times 10^{-3}$ | |
| Liver | Calling | $7.86 \times 10^{-5}$ | $4.30 \times 10^{-4}$ | $4.40 \times 10^{-4}$ | |
| | One-handed | $1.81 \times 10^{-3}$ | $7.91 \times 10^{-4}$ | $2.87 \times 10^{-3}$ | |
| | Two-handed | $6.72 \times 10^{-4}$ | $8.08 \times 10^{-4}$ | $1.06 \times 10^{-3}$ | |

Fig 6 shows that, under combined EMF exposure, the peak whole-body SAR values of the user in three different postures are 0.343 W/kg, 0.193 W/kg, and 0.131 W/kg, respectively, which are 1.06~3.15 times higher than those under single EMF exposure. In particular, during the calling pose, the whole-body SAR value under combined EMF exposure is the highest. The whole-body exposure distribution under combined EMF exposure exhibits a higher SAR than that under single EMF exposure.

Comparison of the SAR distribution in skull tissue is shown in Fig 7.

Fig 7 shows that, under combined EMF exposure, the peak SAR values in skull tissue of the user in three different postures are $2.44 \times 10^{-2}$ W/kg, $2.41 \times 10^{-3}$ W/kg, and $1.53 \times 10^{-3}$ W/kg, respectively, which are 1.38~3.04 times higher than those under single EMF exposure. In particular, during the calling pose, the SAR value of skull tissue under combined EMF exposure is the highest. Under combined EMF exposure, the exposure distribution in skull tissue shows a higher SAR than that under single EMF exposure.

Comparison of the SAR distribution in brain tissue is shown in Fig 8.

Fig 8 shows that, under combined EMF exposure, the peak SAR values in brain tissue of the user in three different postures are $6.82 \times 10^{-3}$ W/kg, $1.34 \times 10^{-3}$ W/kg, and $7.71 \times 10^{-4}$ W/kg, respectively, which are 1.19~2.83 times higher than those under single EMF exposure. In particular, during the calling pose, the SAR value of brain tissue under combined EMF exposure is the highest. Under combined EMF exposure, the exposure distribution in brain tissue shows a higher SAR than that under single EMF exposure, with the exposure concentrated on the side closest to the mobile phone MIMO antenna array.

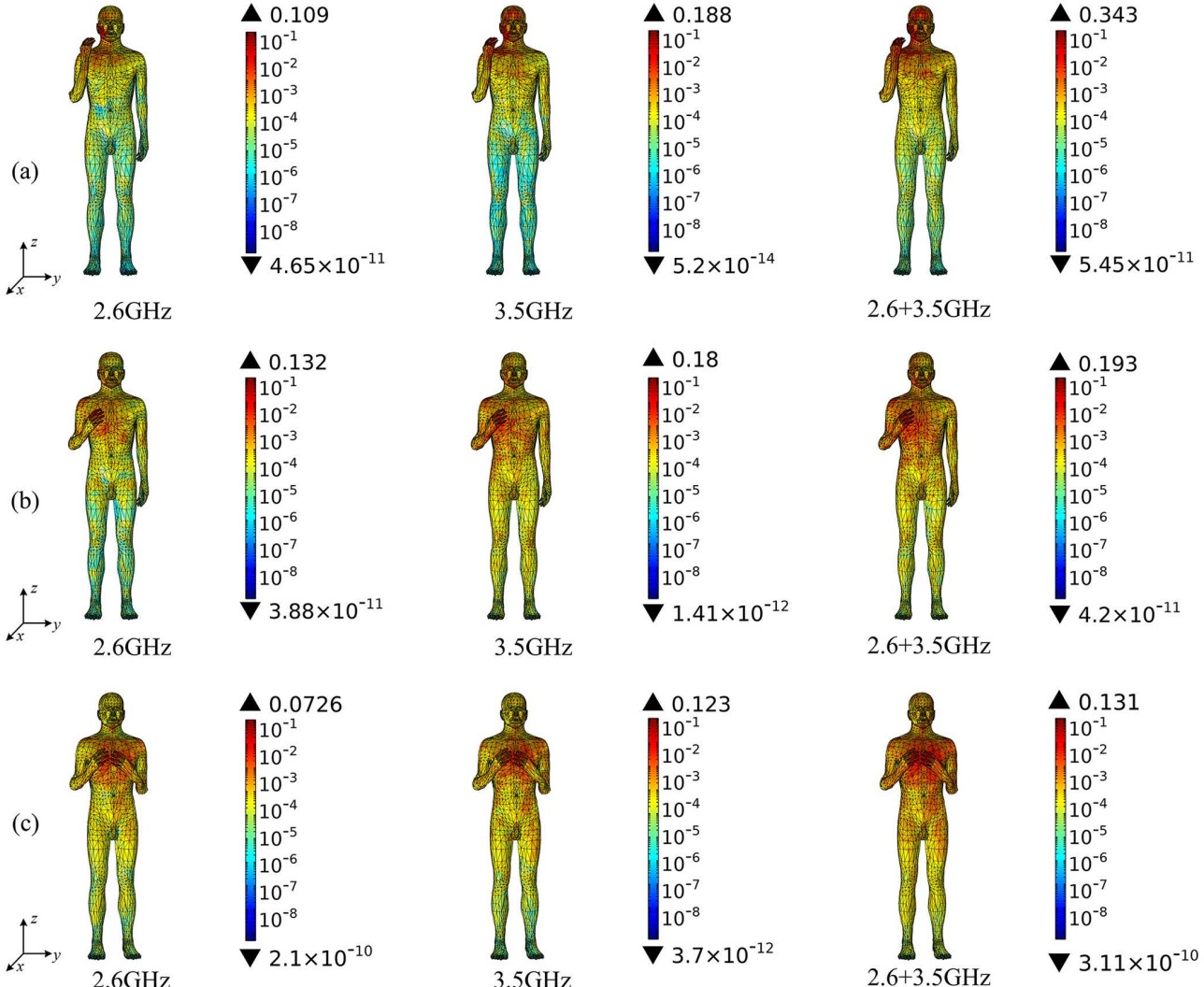

**Fig 6. Comparison of whole-body SAR distributions between single and combined EMF exposure (Unit: W/kg).** (a) Calling pose. (b) One-handed operation pose. (c) Two-handed operation pose.

Comparison of the SAR distribution in thyroid tissue is shown in Fig 9.

Fig 9 shows that, under combined EMF exposure, the peak SAR values in thyroid tissue of the user in three different postures are $1.56 \times 10^{-2}$ W/kg, $2.08 \times 10^{-3}$ W/kg, and $4.57 \times 10^{-4}$ W/kg, respectively, which are $1.01 \sim 5.71$ times higher than those under single EMF exposure. In particular, during the calling pose, the SAR value of thyroid tissue under combined EMF exposure is the highest. Under combined EMF exposure, the exposure distribution in thyroid tissue shows a higher SAR than that under single EMF exposure. During the two-handed operation pose, exposure is primarily concentrated on the lateral aspects of both thyroid lobes, with comparable dose levels.

Comparison of the SAR distribution in heart tissue is shown in Fig 10.

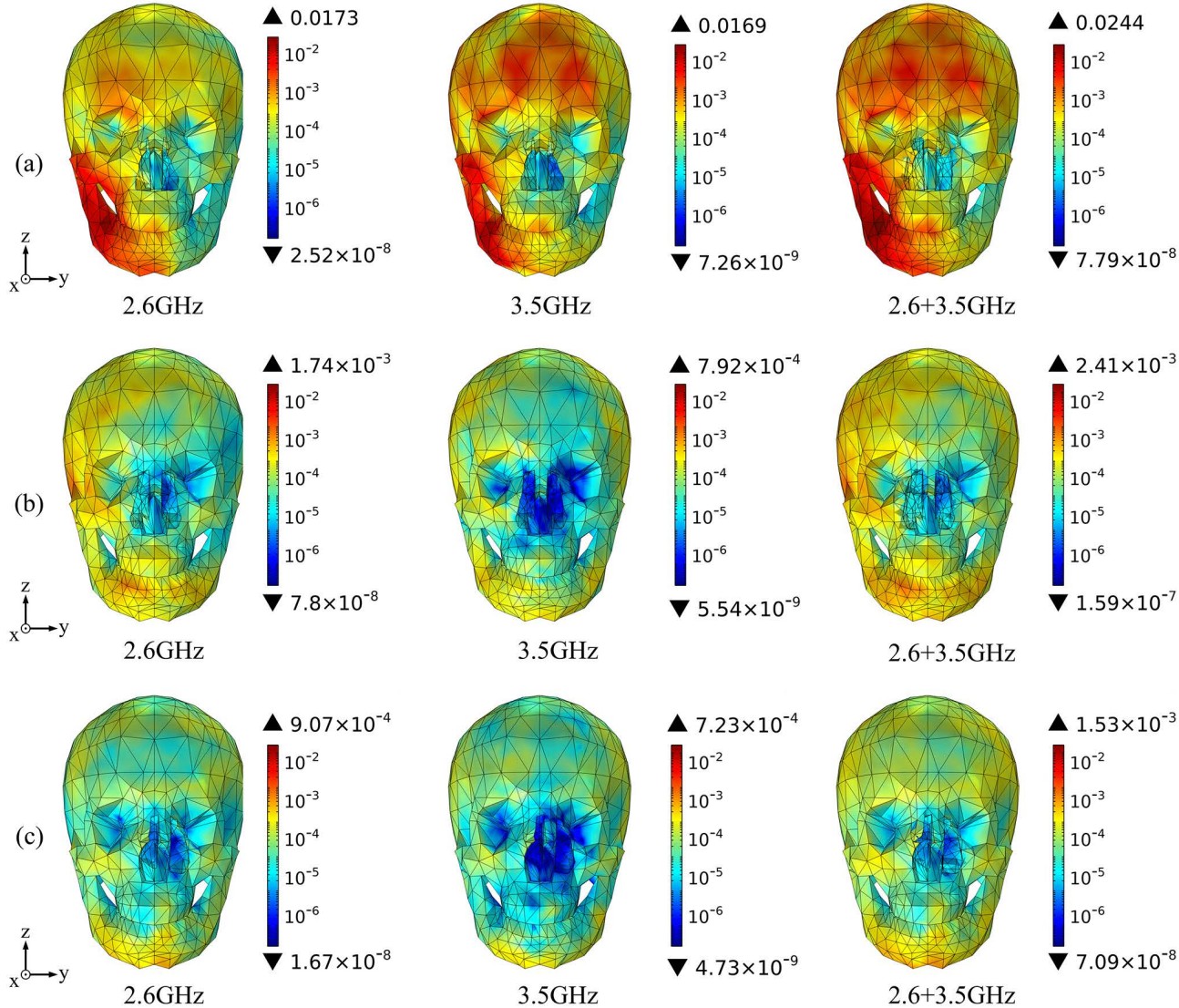

**Fig 7. Comparison of skull tissue SAR distributions between single and combined EMF exposure (Unit: W/kg).** (a) Calling pose. (b) One-handed operation pose. (c) Two-handed operation pose.

Fig 10 shows that, under combined EMF exposure, the peak SAR values in heart tissue of the user in three different postures are $3.67 \times 10^{-5}$ W/kg, $1.77 \times 10^{-4}$ W/kg, and $3.12 \times 10^{-4}$ W/kg, respectively, which are $1.05 \sim 10.72$ times higher than those under single EMF exposure. In particular, during the two-handed operation pose, the SAR value of heart tissue under combined EMF exposure is the highest. Under combined EMF exposure, the exposure distribution in heart tissue shows higher SAR than that under single EMF exposure. Compared to other modes, the heart tissue absorbs the highest radiation dose in the two-handed operation pose.

Comparison of the SAR distribution in lung tissue is shown in Fig 11.

Fig 11 shows that, under combined EMF exposure, the peak SAR values in lung tissue of the user in three different postures are $1.98 \times 10^{-4}$ W/kg, $1.16 \times 10^{-3}$ W/kg, and $1.17 \times 10^{-3}$ W/kg, respectively, which are $1.67 \sim 10.84$ times higher than those under single EMF exposure. In particular, during the two-handed operation pose, the SAR value of lung tissue

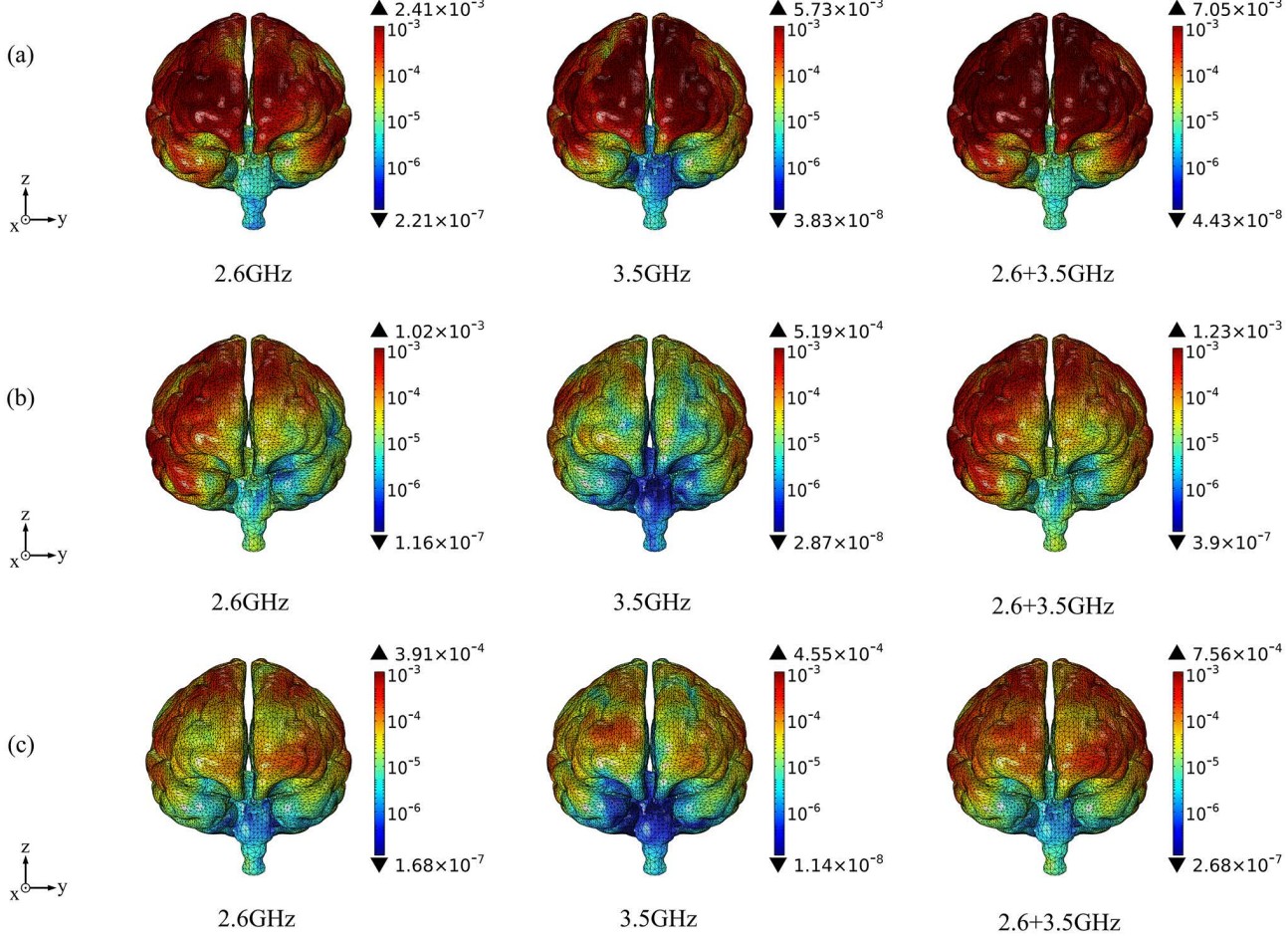

**Fig 8. Comparison of brain tissue SAR distributions between single and combined EMF exposure (Unit: W/kg).** (a) Calling pose. (b) One-handed operation pose. (c) Two-handed operation pose.

under combined EMF exposure is the highest. Under combined EMF exposure, the exposure distribution in lung tissue shows a higher SAR than that under single EMF exposure.

Comparison of the SAR distribution in liver tissue is shown in Fig 12.

Fig 12 shows that, under combined EMF exposure, the peak SAR values in liver tissue of the user in three different postures are $4.40 \times 10^{-4}$ W/kg, $2.87 \times 10^{-3}$ W/kg, and $1.06 \times 10^{-3}$ W/kg, respectively, which are 1.02 to 15.85 times higher than those under single EMF exposure. In particular, during the one-handed operation pose, the SAR value of liver tissue under combined EMF exposure is the highest. Under combined EMF exposure, the exposure distribution in liver tissue shows a higher SAR than that under single EMF exposure.

The above results show that, during the calling pose, the highest SAR levels are observed in the skull, brain, and thyroid tissues, whereas in one-handed operation pose and two-handed operation pose, the highest SAR levels are observed in the heart, lung, and liver tissues. And for each pose, the SAR values in diverse tissues are higher and the exposure distributions are wider under combined EMF exposure compared to the single field. The radiation doses under combined EMF exposure are several times higher than those under single-frequency EMF exposure,

**Fig 9. Comparison of thyroid tissue SAR distributions between single and combined EMF exposure (Unit: W/kg).** (a) Calling pose. (b) One-handed operation pose. (c) Two-handed operation pose.

indicating that such exposure scenarios may lead to a significant increase in radiated energy absorption by various human tissues.

## 5. Conclusion

In this study, we established different scenarios of public mobile phone usage, and explored an active electromagnetic exposure protection approach based on miniaturized radiation source design. Through simulating electromagnetic exposure of a human model in multi-pose, we analyzed the maximum radiation dose absorbed at different frequencies. The findings indicate that during the calling posture the peak SAR values are observed in the head tissues and thyroid tissue. Conversely, in the one-handed and two-handed operation postures, the peak SAR values are present in the thyroid tissue and trunk tissues. As frequency increases, the SAR values in external tissues show a more significant increase compared to those in internal tissues of the human body. The peak SAR values in diverse tissues under combined EMF exposure are 1.02 to 15.85 times higher than those under single

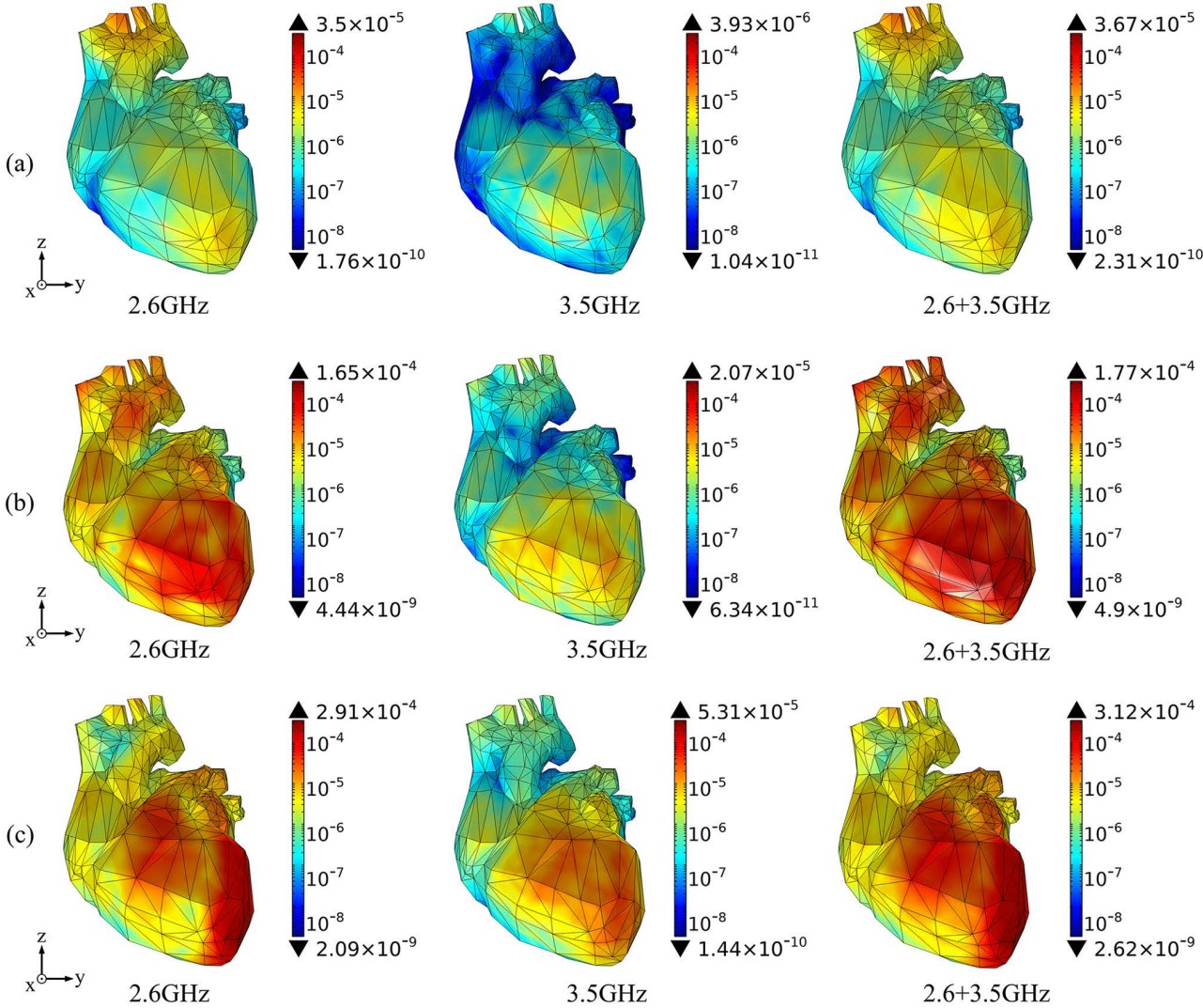

**Fig 10. Comparison of heart tissue SAR distributions between single and combined EMF exposure (Unit: W/kg).** (a) Calling pose. (b) One-handed operation pose. (c) Two-handed operation pose.

EMF exposure. Moreover, in comparison to single EMF exposure, the SAR distributions across diverse tissues demonstrate an overall higher dose level under combined EMF exposure. The uncertainty or sensitivity analysis would be our next work.

## Author contributions

**Conceptualization:** Wen-Ying Zhou.

**Data curation:** Ming-Fei Luo, Wen-Ying Zhou.

**Formal analysis:** Wen-Ying Zhou.

**Funding acquisition:** Wen-Ying Zhou.

**Investigation:** Wen-Ying Zhou.

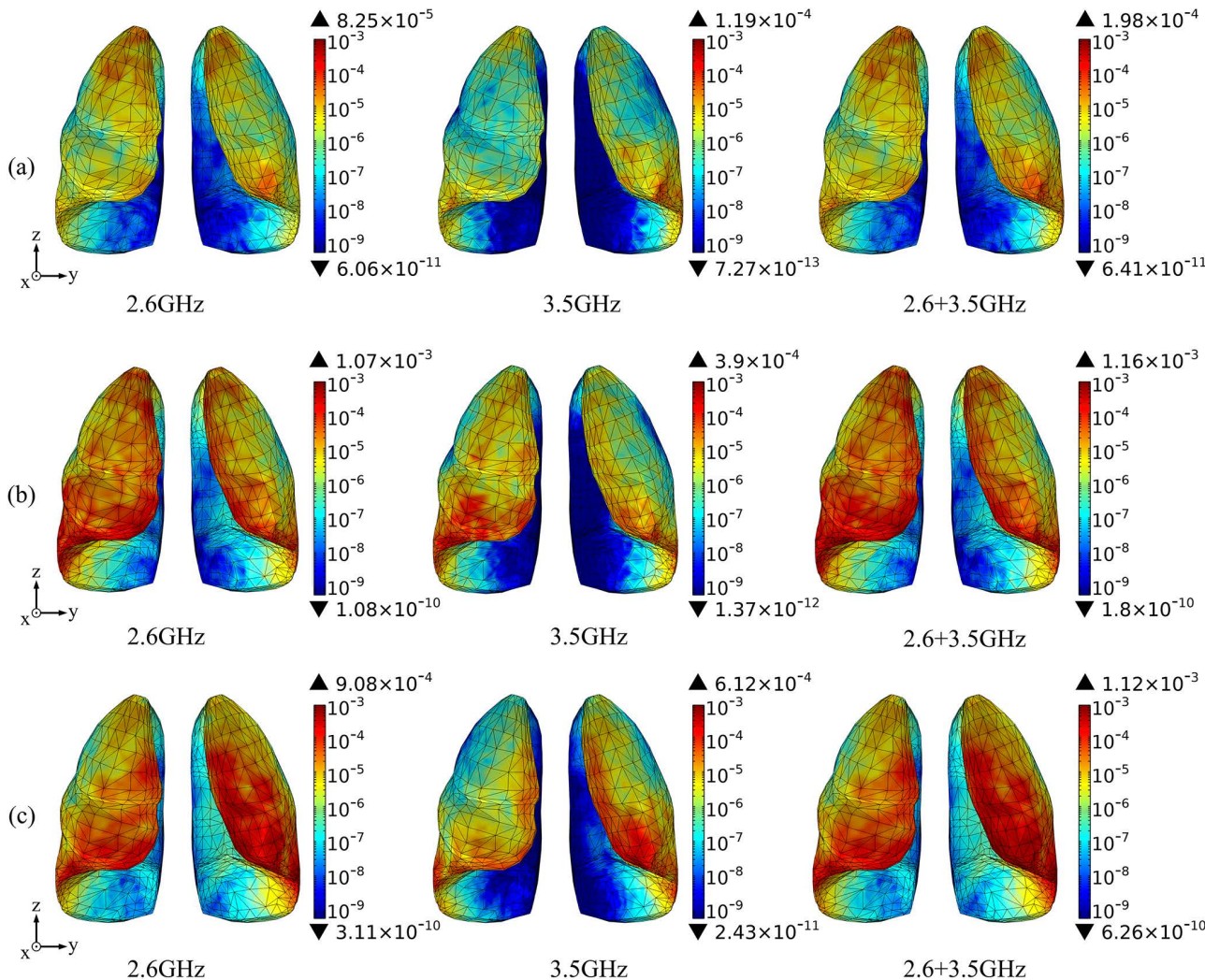

**Fig 11. Comparison of lung tissue SAR distributions between single and combined EMF exposure (Unit: W/kg).** (a) Calling pose. (b) One-handed operation pose. (c) Two-handed operation pose.

**Methodology:** Yu-Xin Li, Wen-Ying Zhou.

**Project administration:** Wen-Ying Zhou.

**Resources:** Wen-Ying Zhou.

**Software:** Yu-Xin Li, Wen-Ying Zhou.

**Supervision:** Wen-Ying Zhou, Mai Lu.

**Validation:** Wen-Ying Zhou.

**Visualization:** Wen-Qi Hou, Yu-Xin Li, Ming-Fei Luo, Wen-Ying Zhou.

**Writing – original draft:** Wen-Qi Hou, Yu-Xin Li, Wen-Ying Zhou.

**Writing – review & editing:** Wen-Qi Hou, Wen-Ying Zhou, Mai Lu.

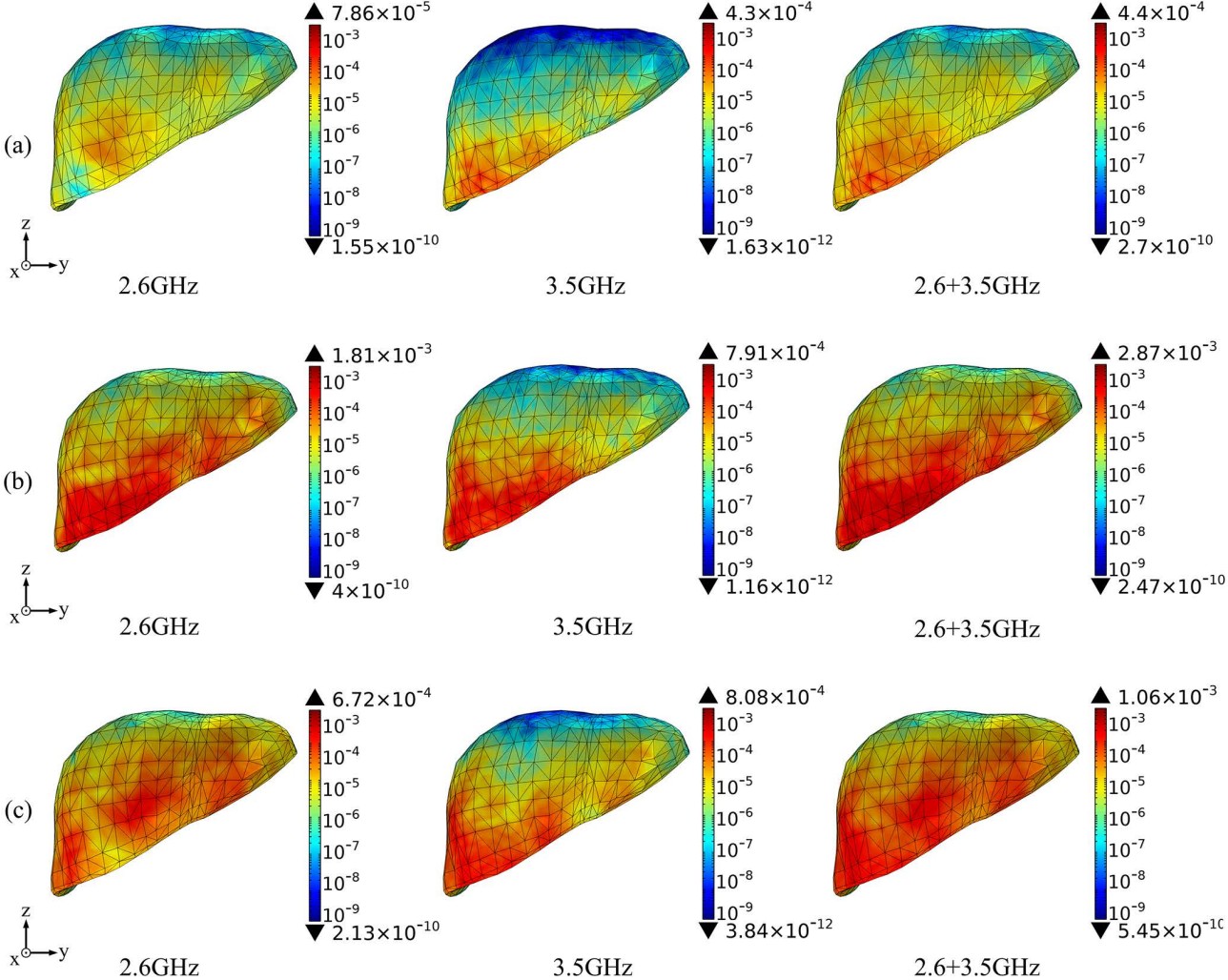

**Fig 12. Comparison of liver tissue SAR distributions between single and combined EMF exposure (Unit: W/kg).** (a) Calling pose. (b) One-handed operation pose. (c) Two-handed operation pose.

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
