## [Decision Letter · Decision Letter 0]

29 Oct 2025

Dear Dr. Hou,

Thank you for submitting your manuscript to PLOS ONE. After careful consideration, we feel that it has merit but does not fully meet PLOS ONE’s publication criteria as it currently stands. Therefore, we invite you to submit a revised version of the manuscript that addresses the points raised during the review process.

We look forward to receiving your revised manuscript.

Kind regards,

Azim Uddin, Ph.D.

Academic Editor

PLOS ONE

Journal Requirements:

“This work was supported by the National Natural Science Foundation of China: [grant number 62161017, 61701208�52467026]; The Department of Education of Gansu Province: [grant number 2024CXPT-11]; The Joint Research Foundation of Gansu Province: [grant number 24JRRA858].”

“This work was supported by the National Natural Science Foundation of China: [grant number 62161017, 61701208�52467026]; The Department of Education of Gansu Province: [grant number 2024CXPT-11]; The Joint Research Foundation of Gansu Province: [grant number 24JRRA858].”

5. We note that your Data Availability Statement is currently as follows: [All relevant data are within the manuscript and its Supporting Information files.]

Reviewers' comments:

Reviewer's Responses to Questions

**Comments to the Author**

1. Is the manuscript technically sound, and do the data support the conclusions?

Reviewer #1: Yes

Reviewer #2: Partly

2. Has the statistical analysis been performed appropriately and rigorously?

Reviewer #1: Yes

Reviewer #2: N/A

3. Have the authors made all data underlying the findings in their manuscript fully available?

Reviewer #1: Yes

Reviewer #2: Yes

4. Is the manuscript presented in an intelligible fashion and written in standard English?

Reviewer #1: Yes

Reviewer #2: No

Reviewer #1: The paper addresses a timely and relevant issue in the context of compound electromagnetic field (EMF) exposure due to the concurrent use of 4G and 5G mobile technologies. The approach of considering simultaneous exposure and the evaluation of miniaturized MIMO antenna arrays introduces a novel aspect to the study, setting it apart from conventional single-frequency SAR analyses. This is a commendable contribution to the field of public exposure safety assessment.

1.The overall structure is logical, but there are several grammatical and syntactical issues that should be corrected for clarity.

2. The SAR distribution figures are informative, but the resolution could be improved. It would also help to include a summary table that clearly compares SAR values across different scenarios (single 4G, single 5G, and compound exposure), and across different body tissues.

3.The use of multi-pose human models adds robustness and realism to the simulation setup. However, the paper would benefit from additional clarification on the following:

(a)How were the poses selected, and do they represent statistically common user behaviors?

(b)Were the MIMO antenna configurations consistent across the 4G and 5G bands, and how were mutual coupling effects between antennas treated in the simulation?

(c)Was the SAR averaged over standard mass values (1g or 10g), and were peak spatial-average SAR values reported in accordance with international guidelines (e.g., ICNIRP, IEEE)?

4.While the paper successfully demonstrates the increased SAR due to compound exposure, it would be valuable to contextualize these findings within the framework of current EMF safety standards. For instance:

(a)Do the maximum SAR values in the compound scenario exceed regulatory limits?

(b)How does the reduction in SAR from the miniaturized MIMO antenna compare quantitatively to conventional antenna designs?

Reviewer #2: The topic is timely and technically relevant for antenna safety design and public EMF exposure assessment, but several major issues in clarity, novelty, and methodologymust be addressed before publication.

The term “compound electromagnetic field” is not standard in EMF research. Please replace it with a more conventional term.

Title can be improved for clarity and conciseness.

The manuscript contains numerous grammatical and technical issues. The entire paper should undergo professional English editing by a native scientific editor.

Please improve figure readability and ensure sequential referencing in text.

Most citations are recent and relevant but not consistently formatted.

The paper builds on the authors’ previous PLOS ONE work (Ref. [20]) but does not clearly distinguish new contributions. Authors should emphasize what is new in this manuscript.

The simulation lacks key details necessary for reproducibility. Author is suggested to add a detailed subsection on simulation details and validation.

Author did not provide any experimental validation. Suitable benchmark SAR studies or validation of model against any standard systems should be included. Author should also include uncertainty or sensitivity analysis.

Methodological description lacks sufficient detail for full reproducibility.

Statements such as “penetration of electromagnetic waves is enhanced under compound EMF exposure” are not physically proved. It is suggested to please revise the discussion to provide consistent explanation.

Figures 6–12 lack clear scale bars and detailed legends. Author is suggested to add scale/color bars showing SAR values and label anatomical regions.

The discussion of Table 5 is minimal. An explanation on why miniaturization reduces SAR would be a helpful addition for the readers.

Some references (e.g., [12]–[14]) need context (which safety standards are meant — IEEE, etc.?).

The inclusion of seven tissues (scalp, skull, brain, thyroid, heart, lungs, liver) is adequate for preliminary SAR analysis, but the model excludes high-exposure regions such as skin, ear, and hand.

Please ensure that units (S/m for conductivity, dimensionless for permittivity) are stated in Table 1.

Conclusion must be improved in term of language and clearer phrasing. Some sentences are grammatically incorrect or ambiguous (e.g., “the organ has affected a lot”), and certain claims such as “penetrability decreases” or “SAR distribution areas expand significantly” should be rephrased more precisely. The authors should also replace wording like “a lot” and “obvious difference.” With scientific wording. Moreoever, SAR increase under compound EMF exposure is not necessarily due to deeper penetration.

It is suggested to rephrase the paragraph of conclusion to form a coherent summary instead of bullet-style listing.

**Do you want your identity to be public for this peer review?** For information about this choice, including consent withdrawal, please see our Privacy Policy

Reviewer #1: No

Reviewer #2: No

---

## [Author Response · Author response to Decision Letter 1]

20 Dec 2025

Dear Reviewers,

Thank you for your comments. We have revised the manuscript and figures accordingly. The title of the manuscript has been adjusted based on your suggestion. The current title is "Mobile phone MIMO antenna array miniaturization-based low SAR research in the compound microwave electromagnetic field", and the original title was "Mobile phone MIMO antenna array miniaturization-based low SAR research in the compound microwave electromagnetic field". Detailed responses to each comment can be found in the file "Response to reviewers."

Best regards.

---

## [Editor Report · Decision Letter 1]

26 Dec 2025

Mobile phone MIMO antenna array miniaturization-based low SAR research in the combined EMF

PONE-D-25-40499R1

Dear Dr. Hou,

We’re pleased to inform you that your manuscript has been judged scientifically suitable for publication and will be formally accepted for publication once it meets all outstanding technical requirements.

Kind regards,

Azim Uddin, Ph.D.

Academic Editor

PLOS One

Additional Editor Comments (optional):

Please modify the manuscript title, the authors wrote EMF, please use the full form in the title.
---

## [Editor Report · Acceptance letter]

PONE-D-25-40499R1

PLOS One

Dear Dr. Hou,

I'm pleased to inform you that your manuscript has been deemed suitable for publication in PLOS One. Congratulations! Your manuscript is now being handed over to our production team.

Kind regards,

on behalf of

Dr. Azim Uddin

Academic Editor

PLOS One